# Management of insecticides for use in disease vector control: Lessons from six countries in Asia and the Middle East

**Henk van den Berg**[1], **Raman Velayudhan**[2], **Rajpal S. Yadav**[2]*

1 Laboratory of Entomology, Wageningen University, Wageningen, the Netherlands, 2 Veterinary Public Health, Vector Control and Environment Unit, Department of Control of Neglected Tropical Diseases, World Health Organization, Geneva, Switzerland

* yadavraj@who.int

**Data Availability Statement:** Detailed country reports were prepared at country level, as explained, and the reports are available from third

## Abstract

Interventions to control the vectors of human diseases, notably malaria, leishmaniasis and dengue, have relied mainly on the action of chemical insecticides. However, concerns have been raised regarding the management of insecticides in vector-borne disease-endemic countries. Our study aimed to analyze how vector control insecticides are managed in selected countries to extract lessons learned.

A qualitative analysis of the situation of vector control insecticides management was conducted in six countries. Multi-stakeholder meetings and key informer interviews were conducted on aspects covering the pesticide lifecycle. Findings were compared and synthesized to extract lessons learned. Centrally executed guidelines and standards on the management of insecticides offered direction and control in most malaria programs, but were largely lacking from decentralized dengue programs, where practices of procurement, application, safety, storage, and disposal were variable between districts. Decentralized programs were better at facilitating participation of stakeholders and local communities and securing financing from local budgets. However, little coordination existed between malaria, visceral leishmaniasis and dengue programs within countries. Entomological capacity was concentrated in malaria programs at central level, while dengue and visceral leishmaniasis programs were missing out on expertise. Monitoring systems for insecticide resistance in malaria vectors were rarely used for dengue or visceral leishmaniasis vectors. Strategies for insecticide resistance management, where present, did not extend across programs or sectors in most countries. Dengue programs in most countries continued to rely on space spraying which, considering the realities on the ground, call for revision of international guidelines.

Vector control programs in the selected countries were confronted with critical shortcomings in the procurement, application, safety measures, storage, and disposal of vector control insecticides, with implications for the efficiency, effectiveness, and safety of vector control. Further international support is needed to assist countries in situation analysis, action planning and development of national guidelines on vector control insecticide management.

parties upon request as detailed in the S2 Appendix.

**Funding:** The work was supported by the Bill & Melinda Gates Foundation, Seattle, WA, as part of the Innovation-2-Impact project (grant number OPP1133139) awarded to the World Health Organization (RV and RSY). The funders had no role in study design, data collection and analysis, decision to publish, or preparation of the manuscript.

**Competing interests:** The authors have declared that no competing interests exist.

## Author summary

Vector-borne diseases such as dengue, malaria and leishmaniasis are transmitted by insect vectors. Transmission can be interrupted through vector control. Chemical insecticides are the mainstay for controlling these insect vectors. However, the use of chemicals also introduces risks to health and the environment and may lead to insecticide resistance. Hence, proper management of those insecticides is critical. To find out how the insecticides used for vector control are being managed, the authors conducted investigations in six countries in Asia and the Middle East. They found that the practices of insecticide procurement, application, storage, and disposal depended on how a program is organized. Dengue programs were operated in a decentralized manner and, consequently, lacked coordination through guidelines and standards on best practices. Also, coordination between malaria, visceral leishmaniasis and dengue programs within countries was minimal, and expertise needed to guide decisions on vector control and to monitor insecticide resistance was in short supply. The identified shortcomings in how vector control insecticides are managed likely affected the efficiency, effectiveness, and safety of vector control operations.

## Introduction

Vector-borne diseases, including malaria, dengue, lymphatic filariasis, schistosomiasis, chikungunya, onchocerciasis, Chagas disease, leishmaniasis, Zika virus disease, yellow fever, Japanese encephalitis and tick-borne diseases, account for an estimated 17% of the global burden of communicable diseases [1,2], and over half of the global population lives in areas at risk of two or more of the major vector-borne diseases [3]. Notwithstanding the diversity of pathogenic agents, their epidemiology, burden and available treatments, all vector-borne diseases have in common that their transmission through mosquito, blackfly, triatomine, sand fly, snail or tick vectors can be interrupted by vector control. Indeed, vector control interventions have demonstrated their critical importance for the control and elimination of most vector-borne diseases [4–8].

Vector control interventions have relied to a large extent on the action of chemical insecticides, applied to surfaces, water bodies, as aerial application, or impregnated in long-lasting netting materials [9–11]. The use of non-chemical options of vector control, such as environmental management or the use of microbial products, remains marginal for a disease like malaria [12], even though the need for alternative methods has been emphasized [4,13–15]. Moreover, the optimal implementation and monitoring of vector control interventions continues to be a major challenge in many countries [16–19].

The World Health Organization (WHO) recently launched the Global Vector Control Response 2017–2030 [1,20] to advance the cause of integrated vector management [21,22]. The strategy is not disease-specific but seeks to strengthen the structures, capacities, and linkages needed to boost the sustainable role of vector control in achieving targets for control and elimination of all major vector-borne diseases. Amongst others, it calls for the improved deployment of available vector control tools, the mainstay of which have been insecticidal tools, by basing selections on efficacy data, with efficient delivery of quality products, with capacity to manage insecticide resistance, and with minimized adverse effects on health and the environment [1].

However, results from a recent study on pesticide lifecycle management showed shortcomings in many countries across regions in terms of insecticide resistance monitoring, procurement procedures, applicator safety measures, storage conditions, and waste disposal [23]. Moreover, regulatory control and quality control of public health pesticides (including vector control insecticides) were a concern, particularly in low-income countries and small countries [23]. These shortcomings in pesticide lifecycle management could compromise the efficient and efficacious use of vector control insecticides, while posing risks to health and the environment.

Detailed studies at country level can provide more insight into the situation of vector control insecticide management, the causes of deficiencies, the context in which decisions have been made, and the opportunities for structural improvements. We conducted an analysis of the situation of vector control insecticides management in selected countries intended to assist these countries in needs assessment and action planning. The objective of this study was to compare and synthesize the analyses from the six countries to extract lessons learned.

## Methods

### Country selection

The selection of countries was conducted based on inquiry from WHO's regional offices (except for the regional office for Europe) about country interest in situation analysis and action planning on public health pesticide management. Following the response from the regional offices, a decision was made to concentrate the study on countries in Asia and the Middle East. After verification of country interest through direct communication with WHO country offices, we verified that interested countries had at least one vector-borne disease control program with a vector control component. Six countries agreed to participate in the activities and all six were selected for study. The selected countries were Bangladesh, Cambodia, Nepal, Oman, Sri Lanka, and Vietnam.

After completion of our data collection, we verified whether the selected countries were representative for their regions. This was possible through examination of the results from a separate global survey on pesticide management, which became available in 2020 [23]. The global survey assessed the country-level situation regarding fifteen quality indicators on vector control insecticide management, covering 31 countries in Asia and the Middle East which obtained an average indicator score of 52% (higher is better). In the same survey, the six individual countries selected for our study had an average indicator score of 40%. This suggests that the selected countries for our study had slightly poorer conditions of vector control insecticide management than other countries in Asia and the Middle East.

### Data collection

Preparatory consultations were held with focal points in each country to identify, and arrange meetings with, the main experts in the management of public health pesticides in each country. A list of the panel of experts in each country, with area of expertise and agency, are available in the Supporting Information (S1 Appendix). Study visits to each selected country were made by one of us (HvdB) over the period August-December 2019. The methods in each country included interview meetings with one or more key staff from each identified stakeholder to gather information and data and, in Bangladesh, Nepal and Oman, a multi-stakeholder meeting, to discuss strengths, weaknesses and needs. Depending on the country, stakeholders included ministries of health, agriculture, local government and environment, vector-borne disease program offices, pesticide registration offices, pesticide analysis laboratories, municipalities, civil society organizations, the private sector, and academic institutions. Visits were

made to pesticide laboratories, entomology laboratories in all countries, and to field spray teams in Bangladesh, Oman, and Sri Lanka. The scope of the assessment included technical, organizational, and institutional aspects of the following topics: pesticide regulatory control, pesticide procurement, insecticide application for vector control, insecticide resistance management, safety measures, pesticide storage, pesticide disposal, and waste management. A list of questions was prepared for each topic as a guide during the interviews, and the responses were documented. Institutional and organizational aspects regarding each topic (e.g. regarding coordination, capacity) were explored through discussion. Data on disease cases were obtained from a national database in each country. For each country we documented the findings in a country report (S2 Appendix). We consulted additional data and information sources, including national guideline or strategic documents, scientific reports, regional workshop reports and published literature to provide verification and insight into the subject matter. Where claims could be verified, we have noted this in the text.

### Data interpretation

The outcomes of the six country reports were reviewed to identify themes of interest. Excluded were issues referring to pesticides other than vector control insecticides. The identified themes were: organizational aspects of vector control, coordination between programs, inter-sectoral coordination, entomological capacity, insecticide resistance management, pesticide regulatory control, pesticide procurement, application methods and safety measures, and pesticide storage and disposal. For each theme, the issues at hand were synthesized to extract relevant lessons learned across countries. Where possible, the information received through verbal communication was verified through review of available documentation and direct observation. Countries were anonymized for information that was considered potentially sensitive.

## Results

### Vector control programs

The number of vector (or nuisance pest) control programs varied from two to four per country (Table 1). Malaria vector control programs existed in all countries, in phases ranging from malaria elimination to prevention-of-reintroduction of malaria. Malaria cases were approaching elimination in Bangladesh, Cambodia, Nepal, and Vietnam; and were zero or almost zero in Sri Lanka and Oman. *Plasmodium vivax* was dominant among malaria cases in Cambodia and Nepal whilst *P. falciparum* was dominant among cases in Bangladesh and Vietnam. Bangladesh and Nepal had vector control programs on elimination of visceral leishmaniasis ('kala-azar'), with relatively small numbers of annual cases reported. The six countries had vector control programs on emergency response and control of dengue, showing a recent increase in the number of cases, particularly in Bangladesh, Cambodia, Sri Lanka, and Vietnam.

In the selected countries, programs on elimination of malaria and visceral leishmaniasis depended primarily on insecticide-treated bed nets and indoor residual spraying for vector control (Table 1), using pyrethroids for both methods, despite partly outdoor-biting behavior of vectors and despite the emergence of pyrethroid resistance [24,25]. Indoor residual spraying was used for reduction of the malaria burden or for emergency control where sporadic malaria cases were found. Insecticidal hammock nets and topical repellents were distributed as additional tools to protect forest-goers against malaria in Cambodia and Vietnam, whilst larviciding was used in receptive sites for prevention-of-reintroduction of malaria in Sri Lanka and Oman.

In dengue control, 'source reduction', which is the removal or coverage of water-holding containers in which dengue vectors breed, was promoted as core intervention in all endemic

**Table 1. The programs with vector control component in the selected countries.**

| Country | Vector-borne disease or pest targeted | Cases[a] | Control stage[b] | Vector control tools[c] | Organizational structure[d] | Actors |
|---|---|---|---|---|---|---|
| Bangladesh | Malaria | 10500 | 2 | 3,4 | C | Malaria program |
| | Visceral leishmaniasis | 144 | 2 | 3,4 | C | Kala-azar program |
| | Dengue | 90000 | 3,4 | 1,2,5 | D | Municipalities, malaria/dengue unit |
| | Nuisance mosquitoes | n/a | 4 | 2,5 | D | Municipalities |
| Cambodia | Malaria | 25502 | 2 | 3,7 | C | Malaria program, PMI |
| | Dengue | 67436 | 3,4 | 1,2,5 | D | Dengue program, provinces, districts |
| Nepal | Malaria | 1065 | 2 | 3,4 | C | Malaria Program |
| | Visceral leishmaniasis | 169 | 2 | 4 | C | Kala-azar program |
| | Dengue | 14000 | 3,4 | 1 | D | Dengue program, municipalities |
| Sri Lanka | Malaria | 1 | 1 | 2,3,4 | C/D | Malaria program |
| | Dengue | 99120 | 3,4 | 1,2,4,5 | D/C | Dengue program, municipalities |
| Oman | Malaria | 0 | 1 | 2,5 | C | Ministry of Health |
| | Dengue | 59 | 3,4 | 2,5,6 | C/D | Ministry of Health, municipalities |
| | Nuisance mosquitoes | n/a | 4 | 2,5 | D | Municipalities |
| Vietnam | Malaria | 5000 | 2 | 3,4,7 | C | Malaria program |
| | Dengue, chikungunya | 250000 | 3,4 | 1,2,5 | D/C | Dengue program, municipalities |

[a]Annual number of cases in 2019 or, where unavailable, in 2018; n/a signifies 'not applicable'

[b]1, Prevention-of-reintroduction; 2, elimination; 3, emergency response; 4, control

[c]1, Source reduction or environmental management; 2, larviciding; 3, long-lasting insecticidal nets (LLIN) or long-lasting insecticidal hammock nets; 4, indoor residual spraying (IRS); 5, space spraying (fogging); 6, peri-focal residual spraying around breeding sites; 7, topical repellents

[d]C, centralized; D, decentralized

countries. Space spraying to kill flying vectors, and larviciding to kill the aquatic immature stages, were targeted at dengue outbreak sites in five countries (Table 1); main insecticides used were the organophosphate temephos for larviciding and pyrethroids for space spraying. Space spraying was conducted mainly outdoors, but in addition in Sri Lanka and Vietnam, ultra-low volume space spraying was applied indoors during an emergency response. In Nepal, space spraying was discouraged by national health authorities because of the notion that spray operations would result in widespread insecticide use and reduce the active role of communities in source reduction activities. In Bangladesh and Oman, space spraying was routinely carried out against nuisance mosquitoes in urban environments.

## Organizational structures for vector control

The way in which vector control is organized at country level has implications for pesticide management. Some programs in the selected countries used a centralized mechanism, in which planning, and financing of vector control were directed by health agencies at national level. Other programs adopted a decentralized or partially decentralized mechanism, in which elements of planning, financing and implementation were delegated to health authorities at the provincial, district or municipal level. Centralized and decentralized programs were operated in parallel within countries.

Centralized programs included those for elimination and prevention-of-reintroduction of malaria, which were generally spearheaded by the national malaria control program authorities–in keeping with the management structure adopted during the first global malaria eradication program [26]. In two countries, components of malaria program management had

been delegated to the provinces and districts. The core vector control interventions (insecticide-treated bed nets and indoor residual spraying) aimed for universal coverage of targeted communities, which may have benefited from centralized planning and resource allocation. Also, epidemiological and entomological surveillance, where available, were mostly planned at the central level. The programs on elimination of visceral leishmaniasis followed a centralized mechanism like for malaria elimination.

Conversely, dengue programs were mostly decentralized to district or municipal health offices. The core intervention, source reduction, depended on the active participation of communities, including schools and workplaces. Hence, source reduction may have benefited from a decentralized setting. Space spraying and larviciding were used in response to dengue outbreaks by district or municipal health offices in five countries. In Bangladesh, dengue outbreak response was taken up by municipal health authorities, tasked to the spray teams responsible for routine control of nuisance mosquitoes. In two countries, the national health authorities assisted smaller municipalities outside of the capital city in their dengue control activities.

Decentralized dengue programs in five countries had a central dengue unit, tasked with strategic development and training. Four countries had developed national guidelines on dengue control. Vietnam established a centralized online reporting system for dengue-case data from hospitals and clinics, which was used by sub-national institutions to advice district health centers on emergency response action. Other countries lacked a national online reporting system and, thus, the central unit did not advice the districts on dengue response action. Financing of dengue control activities, including for training and insecticide procurement, was primarily through budget allocation at district or municipal level according to local priority setting, which reportedly resulted in a variable quality of training and operations. In Vietnam and Sri Lanka, the central level co-financed the districts, by contributing to insecticide procurement or by supporting dengue control.

Centralized mechanisms aided vector control on several fronts. National policies and strategies for the control and elimination of malaria provided direction to program activities. National guidelines, certified training, and a central mechanism for insecticide procurement, were critical to the quality and safety of implementation of vector control and pesticide management but these elements were largely absent from dengue programs. Moreover, in Sri Lanka, centralization served to protect national interests concerning the prevention-of-reintroduction of malaria, a subject likely to be undervalued in decentralized priority setting.

Decentralization supported vector control in other ways. The participation of multiple stakeholders and local communities in the prevention of disease transmission, as envisaged in the Global Vector Control Response, was generally easier to achieve at district or municipal level as compared to the central level, because of shorter communication linkages and closer proximity to the field level. For example, during the 2019-dengue outbreak in Dhaka, Bangladesh, city health departments worked side-by-side with the city's waste management department and with schools to reduce potential breeding sites of *Aedes aegypti* mosquitoes. Conversely, at national level, progress to engage sectors other than the health sector in vector control has been slow or absent in the selected countries. Emergency response action is potentially quicker in a decentralized setting [17], as was found in Colombo Municipal Council, Sri Lanka, where a system of technical experts (epidemiologist, entomologist and pest control officer), case reporting and mobile rapid-response teams reportedly enabled vector control response action within 24 hours of detection of case locations. Furthermore, decentralization improved the prospects for sustained vector control operations because the support for dengue control came from district or municipal budget allocation with little or no external funding. By comparison, centralized programs on malaria elimination in the selected countries faced

the uncertainty of continued support for operations after the external donor funding, on which these programs depended, will terminate (Table 2).

## Coordination between vector control programs

It has been advocated that countries establish coordination between their vector control programs because of potential efficiency gains through the sharing of information, infrastructure and human resources [21]. Oman and Nepal had a national-level vector control unit overseeing all vector-borne diseases. In the case of Oman, this unit coordinated surveillance and outbreak control activities in a situation with only sporadic cases of vector-borne diseases. In Nepal, however, the small national unit with inadequate technical capacity was tasked with coordination of sizable programs for malaria, visceral leishmaniasis, and dengue.

In other countries, separate national programs existed on malaria and dengue, each with their own technical support. In Cambodia and Vietnam, the malaria program and dengue program operated mostly in isolation, except that in Vietnam, the malaria program provided the dengue program with technical support on vector surveillance, insecticide susceptibility testing and efficacy testing. In Sri Lanka, the program for prevention-of-reintroduction of malaria

**Table 2. Lessons and conclusions from the case examples regarding nine themes.**

| | Theme | Lesson or conclusion |
|---|---|---|
| 1 | Organizational structures for vector control | Some program components benefit from centralization whereas other components benefit from decentralization, suggesting that a mix is optimal for improvement of the efficiency, quality, safety, and sustainability of vector control operations. |
| 2 | Coordination between vector control programs | The sharing of expertise, equipment and infrastructure between vector-borne disease programs has clear benefits, particularly in settings with a declining trend in malaria versus an increasing trend in dengue. In general, it is not a good solution to simply merge programs, with the intention to reduce the human resources allocated to them. |
| 3 | Inter-sectoral coordination | Establishing intersectoral collaboration on vector control proved to be difficult, but there are indications that dengue control is a viable entry point, because dengue control depends on the participation of partners at local level, where communication linkages are relatively straightforward. |
| 4 | Entomological capacity | Enhancement of entomological capacity is most urgent in decentralized dengue programs, to guide local decisions on vector control, but the demand for entomologists is not being met by tertiary education systems. |
| 5 | Insecticide resistance management | In malaria control, investments in insecticide resistance monitoring systems have not been accompanied by commensurate support for managing resistance effectively or making alternative products and methods available. In dengue control, a lack of resistance management obstructs the efficacious use of insecticides and poses unnecessary risks to health and the environment. |
| 6 | Pesticide regulatory control | Having a single pesticide registration office has clear advantages for harmonizing procedures and standards across pesticide groups but requires strong linkages between the health and agricultural sectors. |
| 7 | Pesticide procurement | Central coordination over pesticide procurement has been lacking in decentralized dengue control programs, with possible implications for vector control. |
| 8 | Application methods and safety measures | With a variety of actors involved in vector control (programs, districts, municipalities), it is critical to strengthen and harmonize application methods and worker safety precautions through the development of national guidelines and protocols. |
| 9 | Pesticide storage and disposal | Practices of storage and disposal of vector control insecticides in districts and municipalities benefit from national guidelines, standard protocols, training and investment. |

and the dengue program cooperated at the district level, where local malaria teams shared their entomological expertise with recently appointed dengue teams and assisted them in the vector control response to dengue outbreaks. Even though malaria has been eliminated in Sri Lanka, the malaria program intended to maintain its separate structure and capacity for vector control, because merging with the decentralized dengue program would risk that the national importance of prevention-of-reintroduction of malaria would be undervalued in district resource allocation. Bangladesh did not have a separate dengue program but, in response to recent increases in dengue cases, the malaria program had incorporated the control of *Aedes*-transmitted diseases, thus aiding coordination between malaria and dengue.

Hence, two countries had a central coordination unit covering all vector-borne diseases but lacking specific capacity for each disease. Two other countries had separate vector control programs which operated mostly in isolation from each other. In two other countries, the expertise, infrastructure, and equipment available in the malaria program were used to assist the less-resourced dengue program in vector surveillance and control. All countries experienced a declining trend in malaria and visceral leishmaniasis versus an increasing trend in dengue. Hence, it will be vital for countries to ensure continuity of the available entomological expertise and resources at the national level and district level (Table 2).

## Inter-sectoral coordination

Sectors outside the health sector have a potential role in vector control, for example, in pesticide management, drainage or irrigation management, waste management or community mobilization. It has been advocated that countries establish a multi-sectoral committee as first step towards involving other sectors in integrated vector management [1,21]. However, Cambodia and Vietnam did not establish intersectoral linkages or a committee on integrated vector management, whilst Bangladesh and Nepal were planning to establish a committee.

In Oman, earlier attempts in 2007 and 2009 to establish an integrated vector management committee failed, reportedly due to a lack of representation by senior-level decision-makers. In 2017, the committee was re-established, this time with senior-level representation from the implicated ministries, and with a technical task force under its wing. The revived committee demonstrated its value during the 2018-dengue outbreak in Oman when it provided vital leadership to multiple actors, and successfully secured funds from cabinet for emergency response action. Similarly, in Sri Lanka, a multisectoral committee has since been instrumental in the mobilization of communities and stakeholders in source reduction activities for dengue control, but the scope did not extend to malaria vectors (Table 2).

## Entomological capacity

Skilled entomologists (i.e., public health practitioners with basic knowledge about vector biology and epidemiology) are essential for any program with a vector surveillance or vector control component. The number of entomologists ranged from zero to four per vector control program. Nepal had no entomologist in place for any of the vector control programs on malaria, dengue or visceral leishmaniasis, and local technicians were absent under the recently federalized government structure. In other countries, entomologists were mainly positioned within malaria programs, mostly at central offices and with assistance by technicians at subnational or district level. In some countries, however, health authorities expressed concern about the continuity of entomological capacity once malaria has been eliminated.

A critical shortage of entomological capacity was apparent in most dengue programs and in the visceral leishmaniasis programs. In dengue programs, no entomologists were placed at the district or municipal level, which was the level at which decisions were made on insecticide

products, and on the methods, timing and targeting of vector control. An exception was Colombo Municipal Council, Sri Lanka, where an entomologist was present. None of the countries had recently completed a vector control needs assessment [27], but Nepal, Oman and Bangladesh were planning one. Possibilities for tertiary education on medical entomology were absent or limited in the selected countries, suggesting the need for new curricula. Consequently, there was a shortage of graduated entomologists available for recruitment by programs. For example, in Nepal, a vacancy for medical entomologist could reportedly not be filled because qualified candidates were lacking (Table 2).

## Insecticide resistance management

Monitoring to detect evidence of insecticide resistance and management to delay development of insecticide resistance have implications for the registration, procurement, and efficacious use of insecticide products.

Malaria programs in the selected countries had systems in place for monitoring insecticide resistance, using the WHO susceptibility test on *Anopheles* spp. collected from sentinel sites. One exception was Nepal, where monitoring of insecticide susceptibility was not carried out and where no recent data were available.

Different stages of management of insecticide resistance in malaria vectors were evident in the selected countries. In Bangladesh, the malaria program had been using deltamethrin for indoor residual spraying for several years and had not added other insecticide options to its arsenal, despite evidence of resistance in part of the sentinel sites. In Nepal, three pyrethroid insecticides were rotated in three-year cycles, but without being informed by insecticide resistance monitoring. Sri Lanka has had a long history using a rotational and mosaic-patterned use of products of insecticides, based on annual susceptibility data from sentinel sites. However, in Vietnam and Sri Lanka, pyrethroids were the only options for rotations in indoor residual spraying, whilst pyrethroids were also being used in insecticide-treated nets. In most cases, insecticide resistance is expected to develop against all compounds, in this case all pyrethroids, that share a common mode of action [28]. Hence, even where insecticide resistance monitoring was in place, the data were not optimally used to manage managing resistance or to make alternative products available.

Dengue and visceral leishmaniasis programs lacked routine monitoring of insecticide resistance in five countries, because of inadequate technical resources or because the importance of resistance monitoring was under-valued by programs. One exception was Sri Lanka, where susceptibility tests were performed at central and decentralized levels. In Vietnam, the dengue program was supported in susceptibility testing of *Ae. aegypti* by the malaria program, whereas Oman was in the process of including *Ae. aegypti* in routine susceptibility testing. Data available from Bangladesh, Cambodia, Sri Lanka, and Vietnam indicated high to very high levels of resistance in *Ae. aegypti* adults to pyrethroids and, in some cases, to the organophosphate malathion [29–31]. Despite high levels of resistance, dengue programs continued to use pyrethroids for space spraying, due to a lack of acceptable alternative products or methods for emergency control. Cambodia reported widespread resistance in *Ae. aegypti* larvae to the organophosphorus larvicide temephos [32]. In response, a 2019-pilot was conducted with the biological alternative *Bacillus thuringiensis israelensis*.

Nuisance mosquitoes, particularly *Culex quinquefasciatus*, were a main target for routine insecticidal application by municipalities in Bangladesh and Oman. Despite that *Cx. quinquefasciatus* exhibited high levels of resistance to almost all insecticides tested in both countries, routine spraying and product selection remained unaltered. This practice has likely contributed to resistance development in *Ae. aegypti*. The visceral leishmaniasis programs in

Bangladesh and Nepal did not have adequate monitoring or management of insecticide resistance of sandfly vectors in place.

Several countries were in the process of developing strategies for insecticide resistance management in conjunction with the agricultural sector, to coordinate or align the selection and use of insecticides between sectors. In Sri Lanka, malathion and most synthetic pyrethroids had been legally restricted for use in mosquito control, prohibiting their use in agriculture, intended to delay the development of insecticide resistance.

## Pesticide regulatory control

Pesticide legislation, including basic provisions for registration, import, export, manufacturing, procurement, marketing, application, storage, disposal, and enforcement were in place in the selected countries. In four countries, legislation covered all pesticide groups, including those for use in agriculture, livestock, households, and public health. However, in Cambodia and Vietnam, legislation covered agricultural and livestock pesticides, but excluded household and public health pesticides. Consequently, in Cambodia, there was no legislation for household and public health pesticides, while in Vietnam, the health ministry had issued a separate decree on household and public health pesticides.

A single pesticide registration authority covering all pesticide groups, and housed under the agricultural ministry, was in place in Bangladesh, Nepal, Oman, and Sri Lanka. Register offices evaluated the data of submitted products for approval of their sale, use and conditions of use, and were advised by a multi-stakeholder committee, in a process that was harmonized for all pesticides. In Oman, pesticide registration had been initiated in 2014, but was still in transition in 2019, whereby part of the pesticide products available on the market were yet to be registered. Pesticide registration offices had few technical staff in all selected countries, and accordingly, data on risk assessment were not locally generated but obtained from the manufacturer. Two countries had a policy for restricting pesticide products for specific uses. For example, Sri Lanka, restricted most pyrethroids and malathion to public health and carbamates to agriculture.

In Cambodia and Vietnam, the central authority for pesticide registration covered only agricultural and livestock pesticides. In Cambodia, there was no separate registration for public health pesticides. In Vietnam, a separate registration scheme for household and public health pesticides had been established under the health ministry but there was little coordination or harmonization on pesticide registration procedures or standards between the agricultural and health ministries. Non-harmonized pesticide registration may compromise the quality of standards and norms and cause inconsistencies with respect to trade, sale, and use. Hence, a single registration authority for all pesticide groups has advantages (Table 2).

Compliance monitoring and enforcement of pesticide legislation was weak, reportedly with illegal pesticide products widely available in all countries. Moreover, pesticide quality control was a major challenge in all countries. Four countries had reference laboratories for pesticide analysis, but these laboratories were rarely used for analyzing vector control insecticides, whilst their analyses comprised identification of active ingredient but excluded chemical-physical properties and relevant impurities.

## Pesticide procurement

Methods of procurement should safeguard the timely availability of vector control insecticide products of high quality, in appropriate amounts, and with the intended effect on vector populations or pathogen transmission. Hence, pesticide procurement will benefit from having central coordination and guidelines. However, national guidelines on the methods and

procedures of procurement of vector control insecticides were absent from five of the selected countries. All countries required that centralized procurements be restricted to WHO-pre-qualified products [33], but seldomly were the procured consignments of vector control insecticides submitted to quality control.

Malaria and visceral leishmaniasis elimination programs made centralized procurements of insecticides, commonly through a specialized procurement unit within the health ministry. However, procurements for the control of dengue and nuisance mosquitoes were primarily carried out by districts or municipalities, without coordination or harmonization between them, and using local funds. National health authorities had little or no control over the process, products or amounts of procurements by districts or municipalities. This lack of control was expressed as a concern because insecticide selection should be consistent with efforts to manage insecticide resistance, and because uncoordinated procurements could lead to accumulation of expired pesticides (Table 2).

### Application methods and safety measures

The malaria and visceral leishmaniasis programs in the selected countries were supported by national guidelines and standard protocols on insecticide application, including for insecticide mixing, spraying, clean-up, and monitoring. However, the guidelines, protocols and, accordingly, the training curricula, apparently needed updating in at least two countries; for example, it was reported that the quality of training on indoor residual spraying did not meet current international standards. With respect to dengue control, national guidelines and standard protocols on insecticide application were absent from three countries, while no national guidelines were known to be present for control of nuisance mosquitoes (Table 2).

Safety precautions to protect vector control spray workers was a critical shortcoming in programs all six countries. National guidelines or standard protocols for use of personal protective equipment, including gloves, mask, apron, boots, goggles [34], were either absent (4 countries), outdated (1 country) or under development (1 country). Basic training on the use of personal protective equipment by spray workers was commonly in place, but poor availability of quality personal protective equipment was a pressing problem across countries. In dengue programs, and in malaria programs where vector control operations were delegated to local actors, the provision of personal protective equipment for spray teams had to come from local budgets. However, a variable quality and provision of personal protective equipment between districts or municipalities were reported from four countries. We observed in two countries spray workers conducting routine space spraying operations with minimal protection. For space spraying, where droplets are small enough to be inhaled, workers need respiratory protective equipment [35], but in none of the countries, respirators were part of the personal protective equipment in space spraying operations.

A national scheme for health monitoring of vector control spray workers to detect pesticide poisoning (e.g., through blood tests or general health checks for signs and symptoms) was absent in all countries. For example, vector control spray workers in Sri Lanka were advised to get themselves checked regularly, but this was not standardized nor regulated. Moreover, a mechanism for compensation for pesticide poisoning among vector control spray workers was reportedly absent from all countries.

### Pesticide storage and disposal

Insecticides procured by malaria and visceral leishmaniasis elimination programs were stored centrally before being distributed to the provinces and districts for local storage to supply of spray operations on the ground. In addition, in Cambodia, Vietnam and Sri Lanka, the central

stores were used for storing part of the insecticides for dengue control. These central stores were generally reported to be secure and safe for storing pesticides, but we were not able to verify these claims. In one country it was reported that these stores dated back from the first malaria eradication era and were not adequately maintained or modernized. Storage facilities at the provincial, district or municipal level were reportedly of questionable standard in all countries, commonly without a room purpose-made for storing pesticides. Of special concern were dengue control insecticides procured and stored by the districts or municipalities, without involvement or monitoring of the central level (Table 2). In Nepal there was a problem with gradual accumulation of expired pesticides at the central and provincial warehouses, suggesting inadequate coordination on stock management and procurement. Other countries mentioned there was no contemporary problem with accumulation of obsolete vector control insecticides.

None of the countries had a policy or national guideline in place for the safe and environmentally sound disposal of obsolete pesticides and pesticide waste, nor was disposal included as topic in the training for spray teams. In some vector control programs, empty pesticide containers were reportedly left in the field or were punctured to avoid reuse. In other programs, used containers were returned to the storage facilities, but were subsequently buried or burned. There was no facility for high-temperature incineration of pesticide waste in any of the countries. Several past efforts were made by countries, with external donor support, to centralize, safeguard and ship pesticide waste for incineration overseas. Insecticide-treated bed net products were redistributed in countries after several years of use by householders. However, none of the countries had a policy or plan in place for withdrawal and sound disposal of used nets.

## Discussion

Amidst changing epidemiological conditions—notably the elimination and emergence of vector-borne diseases [36,37], countries were confronted with critical shortcomings in the procurement, application, safety measures, storage and disposal of vector control insecticides. These deficiencies in the pesticide 'lifecycle' have implications for the efficiency, effectiveness, and safety of vector control and, thus, for the control and elimination of vector-borne diseases. Vector control was used in centralized programs on malaria and visceral leishmaniasis and, in parallel, in decentralized programs on dengue and other arboviral diseases. The comparison between malaria programs and dengue programs, in particular, provided lessons learned regarding the management of insecticides.

Central-level norms, such as national strategies, guidelines and standards, are vital to the quality and safety of insecticide use for all vector control programs [38], especially at decentralized levels where entomological expertise is scarce or absent [17,39]. These norms provided direction and control at all levels in the malaria and visceral leishmaniasis programs in our study, but were largely lacking from dengue programs, where practices of procurement, application, safety, storage, and disposal were variable between districts. For example, central coordination over pesticide procurement has been lacking in decentralized dengue control programs, with implications for quality control, efficacy of interventions and insecticide resistance management. A consequence of non-harmonized methods is that the quality and safety of operations are compromised. In turn, decentralized programs were better at facilitating the participation of multiple stakeholders and local communities in disease prevention activities and securing vector control financing from local budgets–aspects that centralized malaria programs have struggled with [40,41].

Hence, an optimal vector control program, irrespective of the disease targeted, would have a combination of centralized and decentralized components, using harmonized guidelines and quality control measures while also engaging local stakeholders and communities in the prevention of disease. In this respect, the decentralized dengue program in Vietnam enjoyed support for national guidelines, standards, and training. Similarly, in Sri Lanka, the implementation of vector control for prevention-of-reintroduction of malaria was largely decentralized to the districts and regions, whereas monthly coordination meetings at national level with district-level participation ensured harmonization on risk assessment and insecticide resistance management. The Sri Lankan example suggests how programs that achieved malaria elimination could remain vigilant after donor support has waned by establishing vector control capacity at district and regional level. Nonetheless, national leadership must continue to ensure that adequate resources are allocated to the prevention-of-reintroduction of malaria into the country [42,43], rather than relying exclusively on priority setting at the district level.

Despite the potential synergies between malaria programs and dengue programs, in most instances there was little communication or collaboration between the two programs in individual countries, in terms of sharing of infrastructure, information, expertise and human resources. As entomological capacity was concentrated in malaria programs at central level, the dengue and visceral leishmaniasis programs were largely missing out on expertise needed for their vector control decisions, whilst the demand for entomologists was not met by tertiary education systems. Entomological expertise and capacity are essential for entomological surveillance, monitoring and management of insecticide resistance, and for studying efficacy and effectiveness of existing and new interventions. Therefore, firmer linkages should be established between programs, for optimized use of the limited entomological resources available in countries [39].

An area in which coordination is essential is insecticide resistance management [18,44]. Countries in our study established monitoring systems for insecticide resistance in malaria vectors, often with external donor support, but these systems were rarely used for dengue vectors. Moreover, strategies for insecticide resistance management, where present, did not extend across programs or sectors, except in Sri Lanka. Notably, the routine and excessive use of insecticides against nuisance mosquitoes undermined the efficacy of dengue vector control in urban areas in two countries. Another challenge is that insecticide resistance management within programs was hindered by the scarcity of alternative options with unique modes of action [45]. Several novel vector control products have recently been prequalified by WHO [33]. However, in many elimination settings, only small amounts of insecticides are needed to target remaining transmission foci. The registration and shipment for minor-use products is generally unattractive to pesticide suppliers unless countries join forces on the registration and procurement of these products.

Another subject that warrants special attention is space spraying by districts and municipalities for control of dengue and nuisance mosquitoes. Worldwide, space spraying has been the mainstay response to dengue outbreaks for the past fifty years, despite its critiques asserting it has no impact [46,47]. The research community has paid limited attention to space spraying and, to date, there is no epidemiological evidence that supports the use of space spraying for dengue control [48–53]. In our study, local health authorities continued to use space spraying for dengue control in the absence of evidence on its impact because alternative options for outbreak response were lacking. Also, a main driving force behind the continued use of space spraying was the demand from the public, because the highly visible operations provided the public with a sense of security against diseases and pests. Current international guidelines recommend space spraying for dengue control only in emergency situations to suppress an ongoing epidemic or to prevent an incipient one [54]. These norms were interpreted in several

ways among our study countries, ranging from the restricted use in defined clusters based on efficient case reporting, to the extensive use where dengue cases had spread or where case reporting was deficient. Districts and municipalities continued using those insecticides against which vectors had developed high levels of resistance, and space spraying was mostly conducted outdoors, out of contact with the predominantly indoor-resting *Ae. aegypti* vector. Consequently, space spraying probably had little or no impact on vector populations, let alone on dengue incidence, which implies that valuable resources were wasted [55]. These insights call for a revision of international guidelines to help countries decide about the conditions under which the use of space spraying for dengue control is justified. Guidelines should take into account the 'external costs' of pesticide use [56], including health effects on spray workers in situations where personal protective equipment and health monitoring are inadequate, and the adverse effects on non-target organisms [57,58].

Strengths of our study were that it captured a combination of technical, organizational, and institutional aspects of vector control within countries, thus, offering insight into the contemporary challenges and opportunities. Also, the sample of six countries provided an array of parallels and divergences between countries, enabling us to extract lessons learned. A limitation of our study was, as we mentioned in the beginning of this paper, that the selected countries had slightly poorer conditions of vector control insecticide management than other countries in Asia and the Middle East, suggesting that our findings are moderately generalizable for these regions, but the generalizability to other parts of the World remains unknown. Another limitation of our study was that we acquired much of the information through verbal communication with respondents in each country, while opportunities for independent verification of conditions on the ground were limited.

Can structural improvements be made in the management of vector control insecticides? In the past decade, global policy initiatives on integrated vector management and pesticide management have been frustrated by slow progress at country level [1,23,40,59]. Apparently, the gap between international policy, which called for multifaceted integrated approaches, and the reality on the ground has been too large. Funding support to advance the cause for integrated vector management and pesticide management has been meagre and, disturbingly, has not been substantially raised since WHO launched the Global Vector Control Response in 2017. Where available, international support contributed significantly through a 'technocratic approach', particularly through the establishment of capacity for insecticide resistance monitoring in malaria-endemic countries [23]. However, organizational or institutional aspects have received little attention at country, regional and global level. For example, despite insecticide resistance monitoring being in place, the effective management of insecticide resistance has been obstructed by weak intra- and intersectoral collaboration.

To break the impasse in transitioning towards integrated vector management, we propose that countries start from the bottom up by tackling how the contemporary management of vector control insecticides can be improved. A process of situation analysis, as conducted in our study, necessarily involves technical as well as organizational and institutional aspects. If followed through, this process leads to the required intra- and intersectoral linkages and new options for vector control, as has been envisaged in the Global Vector Control Response. Previous initiatives on integrated vector management were mostly centered around malaria control, however, dengue control can be a more viable entry point for establishing multi-partner participation, because of its decentralized management structure [40,60]. In any case, international support is necessary to assist countries in situation analysis and action planning on vector control insecticide management. Moreover, follow-up technical assistance will be required, as indicated in our study, to support the development of national guidelines and standards on

procurement, application methods, safety measures, storage, and disposal, and to facilitate coordination between programs and sectors.

In conclusion, the deficiencies in the management of the pesticide 'lifecycle', as described in this study, is a neglected subject that has important implications for the control and elimination of vector-borne diseases. Several lessons have been learnt on how to improve organization and coordination of vector control and insecticide management within and between programs and between sectors. Essential to all programs in which insecticides are used is that adequate entomological expertise is available and national-level guidelines and standards on how to manage insecticides are executed. Further efforts are needed to identify specific shortcomings in other countries and to establish an inclusive process of planning to make constructive improvements.

## Supporting information

**S1 Appendix. Panel of experts consulted in the selected countries.**
(DOCX)

**S2 Appendix. List of country reports with availability from third party sources.**
(DOCX)

## Acknowledgments

All persons participating in the stakeholder meetings and interviews in the respective countries are gratefully acknowledged for their valuable inputs. Special thanks are due to Mannan Bengali (deceased), Sabera Sultana, Mya Sapal Ngon, Nguyen Thi Lien Huong, Tran Ahn Dung, Vu Nhat Linh, Dai Tran Cong, Subhash Lakhe, Lungten Wangchuk, Usha Kiran, Manjula Danansuriya, Manonath Marasinghe, Jayakody A. Sumith, Osama Ahmed, Vibol Chan, Luciano Tuseo, Say Chy, and Jean Olivier Guintran, for their inputs and their support in facilitating the meetings during the country missions.

This paper's content is solely the responsibility of the authors and does not necessarily represent the official views of their respective organizations.

## Author Contributions

**Conceptualization:** Henk van den Berg, Raman Velayudhan, Rajpal S. Yadav.

**Data curation:** Henk van den Berg.

**Formal analysis:** Henk van den Berg.

**Funding acquisition:** Raman Velayudhan, Rajpal S. Yadav.

**Investigation:** Henk van den Berg.

**Project administration:** Rajpal S. Yadav.

**Supervision:** Raman Velayudhan, Rajpal S. Yadav.

**Validation:** Henk van den Berg, Rajpal S. Yadav.

**Visualization:** Henk van den Berg.

**Writing – original draft:** Henk van den Berg.

**Writing – review & editing:** Henk van den Berg, Raman Velayudhan, Rajpal S. Yadav.

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
