## [Decision Letter · Decision Letter 0]

26 Jan 2021

Dear %Dr% %Yadav%,

Thank you very much for submitting your manuscript "Management of insecticides for use in disease vector control: Lessons from six countries in Asia and the Middle East" for consideration at PLOS Neglected Tropical Diseases. As with all papers reviewed by the journal, your manuscript was reviewed by members of the editorial board and by several independent reviewers. The reviewers appreciated the attention to an important topic. Based on the reviews, we are likely to accept this manuscript for publication, providing that you modify the manuscript according to the review recommendations. 

Sincerely,

Tiago Donatelli Serafim

Associate Editor

Emily Gurley

Deputy Editor

Points to be addressed:

1) Manuscript will benefit if tick born diseases are also mentioned.

2) Line131: Score% is vague. Authors should detail more.

3)Line 354: "manage managing"?

4) Lines 556-570: Authors should reinstate how they get to the conclusions stated on those sentences.

Reviewer's Responses to Questions

**Key Review Criteria Required for Acceptance?**

**Methods**

-Are the objectives of the study clearly articulated with a clear testable hypothesis stated?

-Is the study design appropriate to address the stated objectives?

-Is the population clearly described and appropriate for the hypothesis being tested?

-Is the sample size sufficient to ensure adequate power to address the hypothesis being tested?

-Were correct statistical analysis used to support conclusions?

-Are there concerns about ethical or regulatory requirements being met?

Reviewer #1: Yes, the objectives were stated properly.

Reviewer #2: Yes

Reviewer #3: Minor revisions.

Reviewer #4: (No Response)

**Results**

-Does the analysis presented match the analysis plan?

-Are the results clearly and completely presented?

-Are the figures (Tables, Images) of sufficient quality for clarity?

Reviewer #1: Results are clear and presented current situations success and failures in vector and disease management

Reviewer #2: Yes

Reviewer #3: Minor revisions. Some references are missing in the tables. Please add them.

Reviewer #4: (No Response)

**Conclusions**

-Are the conclusions supported by the data presented?

-Are the limitations of analysis clearly described?

-Do the authors discuss how these data can be helpful to advance our understanding of the topic under study?

-Is public health relevance addressed?

Reviewer #1: The authors could have suggested more conclusions which will help in better coordination at grass root level, state level, national and international level such as technical expertise role in entomology and epidemiology of diseases. They could suggested measures or bridging gaps of the above for elimination of disease and avoiding re introduction of diseases in respective countries.

Reviewer #2: It will be better if the authors conclude their findings in a succinct manner at the end of the discussion or separately mentioning the limitations

Reviewer #3: The discussion are focused mainly in Dengue and Malaria vector control but not leishmaniasis. Please discuss and contrast the lack of studies in leishmaniasis compared to Dengue and Malaria if needed.

Reviewer #4: (No Response)

**Editorial and Data Presentation Modifications?**

Reviewer #1: (No Response)

Reviewer #2: Some minor edits necessary as indicated in the attached file

Reviewer #3: Minor revision

Reviewer #4: (No Response)

**Summary and General Comments**

Reviewer #1: Authors have well written regarding management of insecticides in selected countries. They could have mentioned in their discussion more about quality control during procurement. In Integrated vector management, checking efficacy of vector control operations like Indoor residual spray is very important to assess the role of vectors and their transmission. Correlation between entomological and epidemiological studies plays an important role in elimination, control, and re-introduction of disease. These studies have to be performed in all countries where vector control and epidemiological interventions are used. As mentioned, technical expertise is required. Entomology experts must be recruited at grass root level and national level also for proper guidance, vector identification and monitoring. This will also help in developing new vector control measures which can be introduced in Integrated vector management. Technical expertise can bridge the gap between international policy and bureaucracy in respective countries. Mainly national experts must play an important role in quality control, identification of resistance and assessing the efficacy of vector control used. Insecticide selection, rotation and mosaic usage has to be suggested according to the current requirement without any delays, which helps in better evaluation, management of disease outbreaks and transmission. The authors should have more focused on the above mentioned which will help in designing global vector control practices. The lessons learned for the studies in different countries should be used in an effective manner for prevention of endemicity in some countries.

Reviewer #2: This manuscript is timely and of high significance for formulating international/global policy for insecticide use management in the countries of different regions of WHO

Reviewer #3: The study titled "Management of insecticides for use in disease vector control: Lessons from six countries in Asia and the Middle East" gathered data from six countries with the goal to to assessed the effectivity of current vector control interventions. The authors showed the shortcomings in current vector control programs in these six countries, and the need for the proper application of current vector control interventions. The authors should discuss more about leishmaniasis programs and not only dengue and malaria programs, this would strengthen the main goal of the manuscript. The manuscript is well written, however, some references are missing and this needs to be corrected.

Reviewer #4: The manuscript by Dr. Yadav et al. brings an extremely relevant assessment of some characteristics of vector control programmes in six different countries. It brings an intersting perspective and raises fundamental issues to the improvement of similar national initiatives across the globe. It is very well designed, conducted, written. I have only very minor text edits (see file).

PLOS authors have the option to publish the peer review history of their article (what does this mean?). If published, this will include your full peer review and any attached files.

Reviewer #1: No

Reviewer #2: No

Reviewer #3: No

Reviewer #4: Yes: Fernando Ariel Genta
---

## [Decision Letter · Decision Letter 1]

31 Mar 2021

Dear %Dr.% %Yadav%,

We are pleased to inform you that your manuscript 'Management of insecticides for use in disease vector control: Lessons from six countries in Asia and the Middle East' has been provisionally accepted for publication in PLOS Neglected Tropical Diseases.

Best regards,

Tiago Donatelli Serafim

Associate Editor

Emily Gurley

Deputy Editor

Reviewer's Responses to Questions

**Key Review Criteria Required for Acceptance?**

**Methods**

-Are the objectives of the study clearly articulated with a clear testable hypothesis stated?

-Is the study design appropriate to address the stated objectives?

-Is the population clearly described and appropriate for the hypothesis being tested?

-Is the sample size sufficient to ensure adequate power to address the hypothesis being tested?

-Were correct statistical analysis used to support conclusions?

-Are there concerns about ethical or regulatory requirements being met?

Reviewer #2: Yes all the objectives of the study have been clearly articulated; the study design is appropriate covering 6 countries of South east Asia.

**Results**

-Does the analysis presented match the analysis plan?

-Are the results clearly and completely presented?

-Are the figures (Tables, Images) of sufficient quality for clarity?

Reviewer #2: Yes results have been well compiled and inferences drawn. The tables and figure showing study countries is appropriate. The figure however is not sharp.

**Conclusions**

-Are the conclusions supported by the data presented?

-Are the limitations of analysis clearly described?

-Do the authors discuss how these data can be helpful to advance our understanding of the topic under study?

-Is public health relevance addressed?

Reviewer #2: Conclusion is well written and is supported by data collated from 6 countries

**Editorial and Data Presentation Modifications?**

Reviewer #2: (No Response)

**Summary and General Comments**

Reviewer #2: Very well written manuscript with critical thought in the final revised version.

PLOS authors have the option to publish the peer review history of their article (what does this mean?). If published, this will include your full peer review and any attached files.

Reviewer #2: No

---

## [Editor Report · Acceptance letter]

27 Apr 2021

Dear Dr. Yadav,

We are delighted to inform you that your manuscript, "Management of insecticides for use in disease vector control: Lessons from six countries in Asia and the Middle East," has been formally accepted for publication in PLOS Neglected Tropical Diseases.

Best regards,

Shaden Kamhawi

co-Editor-in-Chief

Paul Brindley

co-Editor-in-Chief
